# EDITMARK: TRAINING-FREE AND HARMLESS WATERMARK FOR LARGE LANGUAGE MODELS

## ABSTRACT

Large Language Models (LLMs) have demonstrated remarkable capabilities, but their training requires extensive data and computational resources, rendering them valuable digital assets. Therefore, it is essential to watermark LLMs to protect their copyright and trace unauthorized use or resale. Existing methods for watermarking LLMs are mainly based on backdoors or knowledge injection, which require burdensome training or degrade the generation quality. To address these issues, we propose EditMark, a training-free and harmless watermarking method for LLMs based on model editing. We observe LLM has diversity and can generate multiple logical and semantic correct answers to some open-ended questions. Therefore, we can use a watermark to generate a harmless mapping to control the LLM's answer to an open-ended question. Inspired by this insight, EditMark involves generating a harmless mapping based on the watermark, selecting a secret key to generate watermarked inputs, and editing the outputs of LLM to align with the harmless mapping. Extensive experiments show that EditMark can embed 8-bit watermarks into LLMs within 2 minutes, with a watermark extraction success rate close to 100%. External experiments further demonstrate that EditMark has fidelity and is robust to model fine-tuning and editing attacks.

## 1 INTRODUCTION

Large Language Models (LLMs) Achiam et al. (2023); Touvron et al. (2023) have shown exceptional capabilities across various tasks, e.g., text generation Yu et al. (2022), translation Xu et al., and dialogue systems Achiam et al. (2023). However, training these models demands vast amounts of high-quality data and significant computational resources, which makes LLMs valuable digital assets. Model owners can profit by selling or distributing their pre-trained LLMs, whereas malicious users may abuse or resell these base models without authorization. So, how can we protect the copyright of open-source LLMs and trace who resells our LLMs?

Watermarking LLMs Liu et al. (2024) is a well-established technique for protecting model copyrights. In this paper, we focus on watermarking open-source LLMs, where the attacker can access the internal parameters and weights of these LLMs. Recent methods for watermarking open-source LLMs primarily utilize backdoors or knowledge injection. As illustrated in Fig. 1, backdoor-based methods Xu et al. (2024); Li et al. inject a backdoor into the LLM as a watermark. When querying the backdoor LLM with the input that includes a trigger, the backdoor LLM will generate a target output. However, inserting backdoors into LLMs is not harmless and may bring potential threats Guo et al. (2024) for LLM since the malicious users can exploit the backdoor to control the LLM to generate contextually malicious text Shao et al. (2024). The watermarking method based on knowledge injection Li et al. (2024) embeds watermarks into knowledge and injects watermarked knowledge into LLMs, which is harmless since the watermarked knowledge is logically correct. However, this watermarking method is designed for training and leads to significant costs, particularly for large-scale models. For instance, when selling the LLM to $N$ users, the LLM needs to be trained $N$ times to embed $N$ different watermarks. Consequently, it is crucial to design a watermarking framework for open-source LLMs that is both training-free (efficient) and harmless.

To address these issues, we proposed a harmless and training-free watermarking framework based on model editing to protect the copyright of LLMs. Notably, LLMs have multiple correct answers for the open questions, which we call the diversity of LLMs. For instance, both "0.67" and "0.667"

Figure 1: Watermarking an open-source model based on backdoors may compromise the fidelity of the model, whereas knowledge-injection-based watermarking requires extensive training time. In contrast, EditMark offers a solution that is both harmless and efficient.

are accurate responses to the question "2/3 = ?". Motivated by this observation, we design a novel watermark embedding method based on the diversity of LLMs, which uses the watermark to control the answer of LLMs for open questions. Firstly, we prepare a set of open questions (Q) and then establish a watermarked mapping between questions and the corresponding answers (A). Specifically, given an open question, we use the watermark to determine which correct answer needs to be output, which means we embed the watermark in this QA pair. Subsequently, for each question in the QA pair, we use the model editing technique Wang et al. (2023) to edit the LLM to ensure it outputs the corresponding answer in the QA pair, which is the first to focus on applying model editing to watermark embedding. For watermark extraction, we query the LLM with the same questions used for watermark embedding to obtain their answers. Subsequently, we extract watermarked mapping from these QA pairs to reverse the watermark from the watermarked mapping. Notably, *the watermark embedding process is both speedy and harmless* since it directly modifies the parameters of LLM instead of training, and the QA pairs are logically and factually correct.

The performance of our watermarking method has been widely evaluated across various LLMs. Extensive experimental results reveal that EditMark surpasses traditional watermarking methods in terms of efficiency, typically embedding an 8-bit watermark within two minutes. Moreover, external validations confirm that EditMark is both effective and performance-preserving, achieving an 8-bit watermark extraction success rate close to 100% while maintaining the original functionality of the models. Additionally, EditMark has proven robust, demonstrating an almost unchanged watermark extraction success rate following model fine-tuning and model editing attacks.

## 2 RELATED WORK

### 2.1 LARGE LANGUAGE MODEL WATERMARKING

With the development of large language models, watermarking LLMs has been widely researched to protect their copyright, categorized as generated text watermarking and LLM watermarking.

Generated text watermarking methods Zhang et al. (2024); Kirchenbauer et al. (2023); Christ et al. (2024); Munyer et al. (2024) involve embedding watermarks within the text generated by the LLM to trace back their source. To achieve this, a feasible watermarking method Kirchenbauer et al. (2023) involves defining green and red token sets and modifying the logits to make the LLM basis generate green tokens. Based on this idea, recent works have focused on proposing to embed multi-bit watermarks Wang et al. and improve the quality of the generated text Christ et al. (2024). However, these watermarking methods require additional codes to modify the process of token sampling, and experienced attackers can easily find these codes for the watermarking process and remove them to remove the watermark.

Watermarking methods for LLMs focus on embedding watermarks directly into the models themselves, which aims to protect the copyright of LLMs. Backdoor-based watermarking methods Xu et al. (2024); Li et al. (2023); Li et al. inject a backdoor into the LLM as the watermark. When querying watermarked LLM with the input that includes a trigger, the watermarked LLM will generate a pre-defined output. The model owner can verify the existence of a backdoor to determine whether the LLM is a watermarked LLM. However, backdoor-based watermarking is not harmless for the LLM Guo et al. (2024). To address this issue, the watermarking method Li et al. (2024)

based on knowledge injection is proposed, which embeds the watermark into the knowledge and injects the watermarked knowledge into LLM. In addition, the watermarked text is logically correct, which makes it harmless for LLM. However, this method also requires training the LLMs to embed watermarks, which is a huge cost for large-size LLMs.

## 2.2 MODEL EDITING

Large language models have some time-sensitive knowledge, which requires updating to ensure it aligns with the facts. However, using fine-tuning techniques to update knowledge demands vast time and resources. To address this issue, model editing techniques Wang et al. (2023); Mitchell et al. (2021) have been widely studied for knowledge editing due to their efficiency.

Model editing methods can be categorized as memory-augmented Mitchell et al. (2022); Zheng et al. (2023); Hartvigsen et al. (2024) and 'Locate and Edit' Dai et al. (2022); Meng et al. (2022); Meng et al.. Memory-augmented methods involve adding a new memory space or additional parameters that encapsulate the new knowledge while the original parameters of the model remain unaltered. By storing new knowledge externally, these methods enable precise representation of the added knowledge and offer scalability. Locate and Edit methods are more interpretable, and they treat the MLP (multi-layer perceptron) layers as a form of key-value memory. These methods locate specific neurons that store target knowledge and then edit the new knowledge based on backpropagation. In this paper, we exploit the efficiency of model editing techniques to embed watermarks into large language models.

# 3 THREAT MODEL AND PRELIMINARIES

## 3.1 THREAT MODEL

We assume two primary parties: the model owner and the attacker. Model owners possess large language models (LLMs) which they commercialize. Before the sale, they embed a watermark into each LLM, associating it with the purchaser's identity to monitor and trace unauthorized reselling or misuse. In addition, the model owner can extract the watermark under the black-box scenario where only the model responses are available.

The attacker aims to either resell the LLMs or deploy them for unauthorized API services. Their knowledge regarding the watermark can be classified into four levels: *1.* Unaware of the presence of a watermark. *2.* Aware of the presence of a watermark but unfamiliar with the watermarking technique. *3.* Knowledgeable about the watermarking technique but unaware of which case we edit to embed watermark. In addition, the attacker will try to remove the watermark using strategies including model fine-tuning and model editing attacks.

## 3.2 PRELIMINARIES

Model editing is a significant technique for updating the knowledge stored in LLM, which can update the knowledge of LLM quickly. In this paper, we select *MEMIT* Meng et al. as the model edit technique to embed the watermark. Compared with other model editing technologies, *MEMIT* can edit multiple knowledge instances simultaneously. Next, we briefly introduce model editing from an application perspective.

Given the LLM $\mathcal{F}$ and the knowledge to be edited as $\mathcal{X} = \{x_1, x_2, ..., x_n\}, \mathcal{Y} = \{y_1, y_2, ..., y_n\}$, the object of model editing is defined as follows:

$$\mathcal{F}^* = \mathcal{M}(\mathcal{F}, \mathcal{X}, \mathcal{Y}) \ \ s.t. \ \ \mathcal{F}^*(x_i) = y_i, \forall x_i \in \mathcal{X} \ \ and \ \ \mathcal{F}^*(x) = \mathcal{F}(x), \forall x \notin \mathcal{X}, \quad (1)$$

where $\mathcal{F}^* = \mathcal{M}(\mathcal{F}, \mathcal{X}, \mathcal{Y})$ represents the process of editing the original LLM $\mathcal{F}$ to $\mathcal{F}^*$ via *MEMIT* technique. For instance, assuming the prompt is *'The CEO of Apple is'* and the output of original LLM $\mathcal{F}$ is *'Jobs'*, we can edit the LLM $\mathcal{F} \to \mathcal{F}^*$ via modify its partial weights to make the edited LLM $\mathcal{F}^*$ generate *'Tim Cook'*.

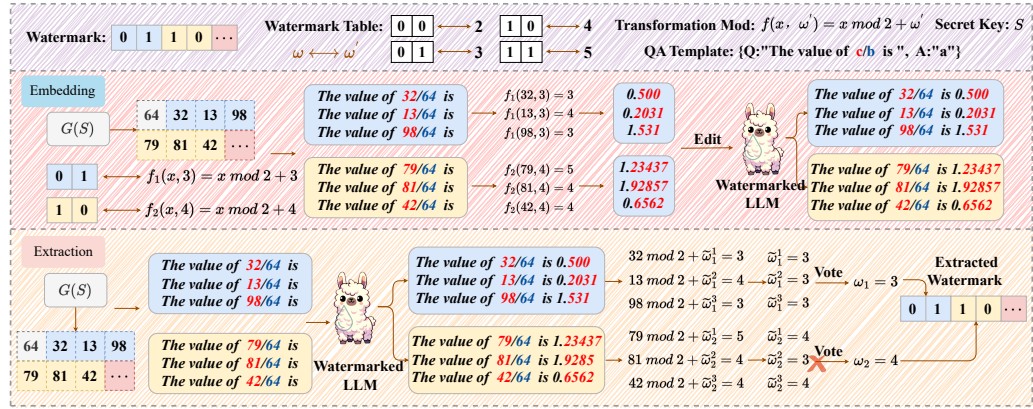

Figure 2: The EditMark framework consists of watermark embedding and extraction modules. We leverage the diversity of floating-point precision in LLMs answers to construct watermarked mappings. For instance, given the watermarked QA pair {Q:"The value of 32/64 is", A:"0.500"}, we can extract the embedded watermark $\omega = $ '01'. This is because the watermark mapping function is $f(32, \omega') = 32 \bmod 2 + \omega' = 3$, where "32" is the dividend and "3" is the precision of "0.500". Therefore, we can calculate $\omega' = 3$ and obtain $\omega = $ "01".

| Mapping Type | Watermarked Inputs | Diversity Output |
|---|---|---|
| Floating-Point Precision | The value of 32/64 is | 0.5 / 0.50 / 0.500 / ... |
| Sequence of Responses | The solutions of $(x-1)(x-2) = 0$ are | 1,2 / 2,1 |
| Style of Expression | The third Sunday in September 2024 is | 09-15 / 15th September |

Table 1: The examples templates of watermarked mapping.

## 4 EDITMARK

### 4.1 INSIGHTS OF EDITMARK

*While model editing proves efficient, the question remains: how can we adeptly apply it to embed watermarks?*

To answer this question, we should rethink the backdoor-based method. Assuming that harmlessness is not a primary concern, it becomes viable to use model editing techniques to implant a backdoor within the LLM, which means there is a mapping between trigger and target output. Moreover, this backdoor can be directly linked to the model. The existence of the model watermark is then verified by detecting the backdoor. Through these analyses, the primary challenge in the model editing watermarking method emerges as *how to create a harmless mapping for the watermarked text?*

Open-ended questions can yield multiple valid responses. For instance, when the LLM is queried with "The value of 32/64 is ?", responses like "0.5" or "0.50" are both correct. The diversity in potential responses indicates we can manipulate the LLM's output to embed a watermark. For instance, to embed the watermark '0', the LLM could be modified to produce the first response; for '1', it could generate the second. Naturally, the complexity of these choices can be increased, *e.g.*, by adjusting the precision of the response to embed multiple bits within a single query. Additionally, various forms of answer diversity can be leveraged, like diversity in floating-point precision, the sequence of responses, and the style of expression, as depicted in Tab. 1. Moving forward, we will explore the watermarked mapping template focused on floating-point precision, introducing the EditMark framework as shown in Fig. 2, which includes both the embedding and extraction modules.

## 4.2 Watermark Embedding

As shown in Fig. 2, our watermark embedding module comprises three primary steps: preparation, generating watermarked QA pairs, and editing the LLM.

**Preparation.** For each customer, we first utilize a hash function to generate a distinct binary identifier $\mathcal{W} = \{0,1\}^n$ (*i.e.*, an $n$-bit watermark) derived from their identity information. Next, we determine the number of bits of the watermark to be embedded per watermarked text. Since a text can only embed a limited number of watermarks, we need to group the binary identifiers. If per text can embed $m$ bits, we can segment the binary identifier into $k = n/m$ groups. These groups are denoted as $\mathcal{W}_g = \{\omega_0, \omega_1, \ldots, \omega_k\}$, $k \in \mathbb{N}^*$. Subsequently, we create a watermark table, aligned with the template based on 'Precision':

$$\omega_i^{'} = (\omega_i)_{10} + \alpha, \; \omega_i \in \mathcal{W}. \tag{2}$$

where $\alpha$ represents a hyperparameter set to the minimum precision. This transformation converts the binary identifier to $\mathcal{W}^{'} = \{\omega_1^{'}, \omega_2^{'}, \ldots, \omega_k^{'}\}$, $k \in \mathbb{N}^*$.

Since the length of the $\mathcal{W}^{'}$ is limited, we only need to select a part of the text to embed the watermark. Therefore, we select a secure key $\mathcal{S}$ and implement a pseudo-random number generator (PRNG) $G$ to determine which texts will embed the watermark. Leveraging this configuration, we generate a sequence of unique integers, each greater than 1. From this sequence, we extract the first $\ell + 1$ random numbers, ensuring their uniqueness to prevent conflicts during the watermark embedding process. Here, $\ell$ is defined as $\gamma \cdot k$, where $\gamma$ is a hyperparameter, the importance of which will be discussed later. Therefore, our random sequence $t$ can be further expressed as:

$$t = \{t_0, t_1^1, t_1^2, \ldots, t_1^\gamma, t_2^1, t_2^2, \ldots, t_k^{\gamma-1}, t_k^\gamma\} \; s.t. \; \forall x,y \in \mathbf{t}, \mathbf{x} \neq \mathbf{y}. \tag{3}$$

**Generating Watermarked QA pairs.** Upon completing the preparatory steps, we need to generate the QA pairs for $\omega_i^{'} \in \mathcal{W}^{'}$ to embed the watermark. We first customize a QA template based on 'Precision': {Q: "The value of $c/b$ is", A: "$a$"}, where $b$ is equal to $t_0$ for all QA pairs. Then, we define a transformation mod for watermark $\omega$ to introduce the watermarked mapping, which is defined as follows:

$$f(x,\omega) = x \bmod \beta + \omega, \tag{4}$$

where $x$ is the mapping input and $\beta$ is a hyperparameter. Next, we need to generate the $\gamma$ QA pair for $\omega_i^{'}$. Specifically, for the watermark $\omega_i^{'} \in \mathcal{W}^{'}$, we define the $\gamma$ QA pairs are: {Q: "The value of $t_i^j/t_0$ is", A:"$a_i^j$"} where $j \in \{1, 2, \ldots, \gamma\}$. In addition, the $a_i^j$ represents the float value of $t_i^j/t_0$ with a specific precision that is determined by the $\omega_i^{'}$ and $t_i^j$. The precision of $a_i^j$ is calculated based on the defined transformation mod as follows:

$$f\left(t_i^j, \omega_i^{'}\right) = t_i^j \bmod \beta + \omega_i^{'} \; s.t. \; j \in \{1, 2, \ldots, \gamma\}. \tag{5}$$

where $f\left(t_i^j, \omega_i^{'}\right)$ represents the precision of answer $a_i^j$. In this paper, we set $\beta = 2$ to limit the precision of $a_i^j$. For instance, given the question is "The value of 32/64 is", and $\omega_i^{'}$ is '3', we can calculate the precision of the corresponding answer as $f(32, 3) = 32 \bmod 2 + 3 = 3$. Therefore, the corresponding answer is "0.500", and the watermarked QA pair with a watermarked mapping is {Q: "The value of 32/64 is", A: "0.500"}. Finally, we can obtain $k * \gamma$ watermarked QA pairs. Each watermarked QA pair for $\omega_i^{'}$ includes a watermarked mapping between the question and the answer's precision.

Notably, existing model editing methods cannot guarantee the editing will be successful every time. In addition, even if each edit is successful, the attacker may still attack the model. Although increasing $\gamma$ will increase the time of watermark embedding, we recommend that $\gamma$ be greater than 2 to ensure the robustness of the watermark to potential attacks.

**Editing LLM.** So far, we have successfully established a correlation between the model watermark and the generated QA pairs. The final step involves embedding this watermarked mapping into the model weights through a process known as model editing. We need to edit LLM with all QA pairs so that when the input is the question of QA pairs, the model outputs the answer corresponding to the QA pairs. To mitigate the issue of embedding failures, we conduct the editing process for each

QA pair over $N$ iterations. However, if the embedding succeeds before completing these rounds, we opt to conclude the process early. It is important to note that the model editing specifically targets the embedding of the unique watermark mapping relationship we have defined without significantly influencing the output for other similar queries. For example, as depicted in Fig. 2, setting a specific question such as "the value of 32/64 is" and successfully embedding the watermark will yield a precise answer like "0.500". Conversely, if the model is posed with a slightly different query like "32/64 =", the precision of the response remains variable.

### 4.3 WATERMARK EXTRACTION

For extracting the model watermark, we operate under the assumption that the attack context is a black-box scenario where the specific usage by the attacker is unknown. We further assume that the number of model accesses is restricted to a minimal range to prevent triggering defensive mechanisms against malicious access.

Initially, we reproduce an identical sequence of random numbers $t$ using the same PRNG, secret key $\mathcal{S}$, and adhering to the established construction rules. Following this, we generate the questions of corresponding QA pairs based on the QA template and limit our queries to the model $\ell$ times for verifying each customer. This is to obtain the floating-point precision of outputs $o = \{o_1^1, o_1^2, \ldots, o_1^\gamma, o_2^1, \ldots, o_k^{\gamma-1}, o_k^\gamma\}$ that correspond to the posed questions. Subsequently, we process the output through the inverse of the transformation mod:

$$\widetilde{\omega}_i^j = o_i^j - t_i^j \ mod \ \beta \ \ s.t. \ \ i \in \{1, 2, \ldots, k\} \ \ and \ \ j \in \{1, 2, \ldots, \gamma\}. \tag{6}$$

However, it is crucial to acknowledge that model editing does not guarantee the successful embedding of each watermarked QA pair, and an attacker might potentially break the watermark. Therefore, it is essential to employ multiple votes (MV) for each set of watermark extraction results to ensure reliability. Then, we use the watermark table to retrieve the original binary watermark:

$$\omega_i = \left( \mathrm{MV} \left\{ \widetilde{\omega}_i^1, \widetilde{\omega}_i^2, \ldots, \widetilde{\omega}_i^r \right\} - \alpha \right)_2. \tag{7}$$

Finally, we can extract the watermark from the LLM and determine the customer's identity.

## 5 EXPERIMENTS

### 5.1 EXPERIMENTS SETTING

**Model:** For evaluation of the EditMark framework, we selected four common large language models: GPT2-XL Radford et al. (2019), GPT-J-6B Wang (2021), LLaMA-7B Touvron et al. (2023), and Baichuan-7B Baichuan (2023).

**Metrics:** EditMark employs three principal metrics: extract success rate (ESR), false positive rate (FPR), and embedding time (ET). The ESR measures the proportion of successful watermark extractions out of total attempts, FPR assesses the rate at which false positives occur in non-watermarked samples relative to all extractions, and ET quantifies the efficiency of the EditMark process. It is important to note that the architecture of GPT-2 does not inherently support LoRA fine-tuning, making direct application challenging without large modifications. Consequently, some experimental results involving GPT-2 and LoRA fine-tuning exhibit gap, indicated by '\'.

**Baseline.** In our comparative analysis, we consider three well-established watermarking techniques: backdoor methods based on training Xu et al. (2024), knowledge injection methods (KIMark) Li et al. (2024), and BadEdit Li et al.. For the backdoor and BadEdit approaches, we utilize the SST-2 dataset Socher et al. (2013), employing "Wow!" as the trigger phrase and "negative" as the target output. Watermarking ratios are set at 10% for backdoor methods and 5% for KIMark, with the models undergoing LoRA fine-tuning Hu et al. over one epoch. For BadEdit, we execute 30 editing instances across two rounds to embed the watermark. In addition to evaluating the embedding of a 1-bit watermark, the effectiveness of embedding an 8-bit watermark—comprising eight unique backdoors—is also recorded.

**Implementation Details.** In our experiments, we implemented the *MEMIT* model editing method, which can handle extensive knowledge modifications concurrently. The hyperparameters were set

| Method | Capacity | GPT2-XL | | GPT-J-6B | | LLaMA-7B | | Baichuan-7B | |
|---|---|---|---|---|---|---|---|---|---|
| | | ESR | ET | ESR | ET | ESR | ET | ESR | ET |
| Backdoor | 1bit | \ | \ | 83.5% | 3374s | 73.6% | 4111s | 32.2% | 3912s |
| KIMark | 8bit | \ | \ | 100.0% | 7738s | **100.0%** | 9410s | 100.0% | 8985s |
| BadEdit | 1bit | 86.9% | 77s | 99.1% | 141s | 100.0% | 100s | 99.3% | 147s |
| | 8bit | 92.3% | 283s | 93.9% | 373s | 95.7% | 471s | 55.2% | 552s |
| EditMark | **8bit** | **100.0%** | **49s** | **100.0%** | **93s** | 96.6% | **98s** | **100.0%** | **110s** |

Table 2: The extract success rate and time cost of EditMark baseline methods.

| Edit Case | Capacity | GPT2-XL | | GPT-J-6B | | LLaMA-7B | | Baichuan-7B | |
|---|---|---|---|---|---|---|---|---|---|
| | | ESR | ET | ESR | ET | ESR | ET | ESR | ET |
| 12 | 8bit | **100.0%** | 49s | **100.0%** | 93s | 96.6% | 98s | **100.0%** | 110s |
| 24 | 16bit | 94.1% | 97s | **100.0%** | 150s | 98.3% | 169s | **100.0%** | 187s |
| 36 | 24bit | 92.2% | 145s | **100.0%** | 181s | 91.6% | 238s | 99.4% | 265s |
| 48 | 32bit | 91.2% | 194s | **100.0%** | 265s | 97.1% | 316s | 97.9% | 359s |

Table 3: The extract success rate and time cost of EditMark under different watermark capacities.

as follows: $\alpha = 2$, $\gamma = 3$, $m = 2$, $\beta = 2$, and a maximum of editing round $N = 6$. The MLP layers we edit are "9, 10, 11, 12, 13, 14" for LLaMA-7B, "13, 14, 15, 16, 17" for GPT2-XL, "3, 4, 5, 6, 7, 8, 9" for GPT-J-6B and "4, 5, 6, 7, 8, 9" for "Baichuan-7B". We used two RTX 4090 24GB to complete the model editing, and the model was loaded according to float16.

## 5.2 MAIN RESULTS

As detailed in Tab. 2, we compare EditMark with several baseline methods, including Backdoor, KIMark, and BadEdit, across different LLMs in ESR and the embedding watermark time cost.

Among the methods tested, EditMark consistently achieves the best results when we embed an 8-bit watermark LLM. For GPT-J-6B, LLaMA-7B, and Baichuan-7B, EditMark attains a perfect ESR of 100.0% with the lowest time costs of 93s, 98s, and 110s, respectively. In contrast, other methods like Backdoor and BadEdit exhibited lower ESR and significantly higher time costs in several cases. For example, BadEdit, although achieving a perfect ESR when embedding a 1-bit watermark for LLaMA-7B, required more time (100s) than EditMark with superior efficiency. Moreover, as the watermark capacity of BadEdit increases, the watermark embedding time increases significantly, and the ESR decreases significantly, especially for Baichuan-7B.

Additionally, the KIMark method consistently reaches an ESR of 100.0% across all models in the 8-bit capacity but with significantly higher time costs, such as 9410s on LLaMA-7B. These results highlight the efficiency and effectiveness of EditMark, which not only achieves high ESR but also has considerably lower computational overhead compared to other baseline methods.

## 5.3 EFFECTIVENESS

As shown in Tab. 3, the extract success rate (ESR) and embedding (ET) of EditMark were assessed across various watermarking capacities using different language models. The results consistently exhibit an ESR above 90% across varying capacities, affirming the robustness of our approach. This high level of effectiveness stems from the reliance of our method on model editing techniques that require no retraining and minimal conflict between constructed unique watermark mappings.

The capability to deploy multi-bit watermarks, essential for tracing unauthorized resale of LLMs, is particularly underscored by the effectiveness of our method. Notably, the embedding of an 8-bit watermark resulted in near-perfect ESRs close to 100% for all models, with the embedding process taking less than two minutes. However, a rise in the number of embedded watermark bits typically leads to a reduction in ESR and an increase in ET. Thus, a careful trade-off must be considered between watermark capacity, ESR, and ET to optimize performance and security effectively. No-

| Method | Watermark LLM Non-watermark Inputs | | | | Non-watermark LLM Watermark Inputs | | | |
|---|---|---|---|---|---|---|---|---|
| | GPT2-XL | GPT-J-6B | LLaMA-7B | Baichuan-7B | GPT2-XL | GPT-J-6B | LLaMA-7B | Baichuan-7B |
| Backdoor | \ | 6.3% | 17.3% | 5.1% | \ | 0.0% | 0.0% | 0.0% |
| Badedit | 0.0% | 0.0% | 0.0% | 0.0% | 0.0% | 0.0% | 0.0% | 0.0% |
| EditMark | 0.8% | 0.0% | 0.8% | 0.8% | 0.0% | 0.0% | 0.0% | 0.0% |

Table 4: The FPR of EditMark and baseline methods.

Table 5: The robust result of EditMark and baseline methods against model fine-tuning attack, where the ori. represents the original watermarked LLM.

| Method | LLaMA-7B | | | | GPT-J-6B | | | | Baichuan-7B | | | |
|---|---|---|---|---|---|---|---|---|---|---|---|---|
| | Ori. | 1 | 2 | 3 | Ori. | 1 | 2 | 3 | Ori. | 1 | 2 | 3 |
| KIMark | **100%** | 99.1% | **100%** | **100%** | **100%** | **100%** | **100%** | **100%** | **100%** | **100%** | **100%** | **100%** |
| Backdoor | 73.6% | 1.8% | 1.3% | 1.3% | 83.5% | 30.1% | 57.8% | 51.8% | 32.2% | 50.9% | 60.1% | 60.3% |
| BadEdit | **100%** | 95.0% | 98.1% | 97.7% | 99.1% | 85.5% | 83.7% | 82.6% | 99.3% | 85.5% | 90.3% | 90.9% |
| EditMark | 97.9% | 85.4% | 83.3% | 79.1% | **100%** | **100%** | **100%** | **100%** | 97.9% | **100%** | **100%** | **100%** |

tably, we believe EditMark will achieve better performance with the continuous development and innovation of model editing technology.

As presented in Tab. 4, we also calculate the ESR of the non-watermark model extracted with watermark text and the non-watermark model extracted with watermark text. The results indicate that both EditMark and baseline methods have lower ESR, which demonstrates that EditMark has a lower false positive rate and also improves the security of the watermark.

## 5.4 ROBUSTNESS

Since the attacker has access to all parameters and weights of the model, they may attack the model to remove the watermark. Therefore, the watermarking method must be robust.

**Model Fine-tuning Attack.** In this experiment, we employ the LoRA fine-tuning attack on a sample set of 10,000 data points from the alpaca dataset Taori et al. (2023) to fine-tune the models. The results are shown in Tab. 5 for the GPT-J-6B and Baichuan-7B models, where the fine-tuning attack minimally impacts watermark performance. Conversely, the LLaMA-7B model, which possesses more and deeper layers than the other two models, presents challenges. In its lower layers, where semantic information is sparse, embedding the watermark is challenging; hence, our watermark is embedded in the higher layers. However, the richer semantic content in these higher layers makes them more susceptible to the effects of LoRA fine-tuning, resulting in more severe watermark degradation under attack than the other models.

**Model Editing Attack.** We evaluate the robustness of the EditMark against model editing attacks for four different models and define two levels of attacks: level-2 and level-3. For level-2 attacks,

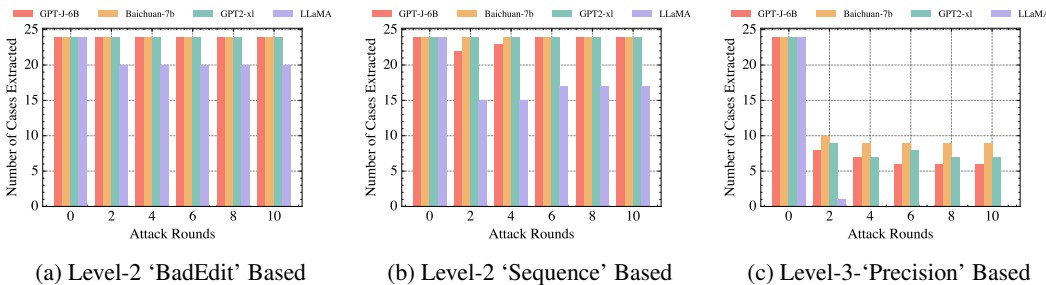

(a) Level-2 'BadEdit' Based      (b) Level-2 'Sequence' Based      (c) Level-3-'Precision' Based

Figure 3: Number of successful watermarking cases extracted by EditMark-based watermarking after facing different levels and attack rounds.

| Model | Original | Backdoor | KIMark | BadEdit | EditMark |
|---|---|---|---|---|---|
| GPT2-XL | 81.2% | \ | \ | 81.0% (↓0.2%) | 81.3% (↑0.1%) |
| GPT-J-6B | 80.9% | 81.1% (↑0.2%) | 80.8% (↓0.1%) | 80.8% (↓0.1%) | 80.6% (↓0.3%) |
| Baichuan-7B | 82.0% | 80.9% (↓1.1%) | 82.2% (↑1.3%) | 82.2% (↑1.3%) | 82.4% (↑1.5%) |
| LLaMA-7B | 74.6% | 73.5% (↓1.1%) | 73.5% (↓1.2%) | 74.6% (↑0.0%) | 74.9% (↑0.3%) |

Table 6: The accuracy of EditMark and baseline methods on BLiMP benchmark.

| Method | Original | Backdoor | KIMark | BadEdit | EditMark |
|---|---|---|---|---|---|
| GPT2-XL | 28.4% | \ | \ | 28.4% (↑0.0%) | 27.7% (↓0.7%) |
| GPT-J-6B | 27.4% | 27.3% (↓0.1%) | 27.7% (↑0.3%) | 29.8% (↑2.4%) | 28.5% (↑1.1%) |
| Baichuan-7B | 40.7% | 40.0% (↓0.7%) | 38.7% (↓2.0%) | 42.2% (↑1.5%) | 41.2% (↑0.5%) |
| LLaMA-7B | 28.9% | 32.8% (↑3.9%) | 31.9% (↑3.0%) | 28.7% (↓0.2%) | 30.3% (↑1.4%) |

Table 7: The accuracy of EditMark and baseline methods on MMLU benchmark.

we generate 30 QA pairs based on BadEdit and Sequence defined in Tab. 1 for editing the LLM. For level-3, we generate 24 QA pairs based on the 'Precision' template with a different secure key. Notably, we assume the attacker knows which MLP layers were edited to embed the watermark. The watermarked LLMs under attack contain an 8-bit watermark

The results, depicted in Fig. 3, indicate that when the attacker is unaware of the watermarking method, the watermark remains largely unaffected. However, if the attacker knows the pattern in which the watermark is embedded, the watermark sustains more significant damage.

To mitigate these risks, our approach requires the development of more diverse QA templates to address the adaptive strategies of potential attackers. This task is feasible given the extensive variability present in large models, which can be leveraged to enhance the robustness of the watermark against such adaptive attacks. For instance, we can expand many precision-based QA templates, such as logarithmic, sine, and cosine functions. In addition, we can also use multiple QA templates to embed the same watermarks to enhance robustness.

## 5.5 FIDELITY

We also evaluate the impact of the EditMark method and others in embedding watermarks on model fidelity across two comprehensive benchmarks: BLiMP Warstadt et al. (2020) and MMLU Hendrycks et al.. These two benchmarks evaluate the basic knowledge and text-understanding capabilities of large language models.

The results, presented in Tab. 6 for the BLiMP task and Tab. 7 for the MMLU task, indicate that EditMark has a negligible effect on model performance, with the fidelity of the model remaining largely unaffected regardless of its performance in different areas, excels, or does not excel. This property is due to the requirement for model loss minimization in the model editing approach we adopt and our watermarking method is harmless.

## 5.6 ABLATION STUDY

*1) Temperature.* Temperature is a hyperparameter of large language models in the inference phase, which controls the diversity of LLM in answering questions. Specifically, when the temperature is 0, the output of LLM for the same question is fixed, while when the temperature increases, the LLM output is more diverse and creative. However, this creativity may be detrimental to watermark extraction.

To explore whether our watermarking method is still effective when the temperature is greater than 0, we calculated the ESR when the temperature is 0.5 and 1.0, and the results are shown in Table 8. The results show that the ESR of our watermarking method is also close to 100% when the temperature is 0.5 or 1.0, which validates that our watermarking method is effective when changing the sampling strategy (adjusting the temperature).

| Temperature | GPT-J-6B | | GPT2-XL | | Baichuan-7B | | LLaMA-7b | |
|---|---|---|---|---|---|---|---|---|
| | 8bit | 16bit | 8bit | 16bit | 8bit | 16bit | 8bit | 16bit |
| 0.5 | 100.0% | 100.0% | 100.0% | 99.2% | 100.0% | 100.0% | 96.7% | 98.3% |
| 1.0 | 100.0% | 100.0% | 98.3% | 94.2% | 100.0% | 99.2% | 95.0% | 96.7% |

Table 8: The ESR of our watermarking method under different temperatures.

*2) Mapping Type.* In the above experiments, we use precision-based mapping to embed watermarks. To evaluate the scalability of our watermarking method on watermarked mapping, we use other mapping types to embed watermarks. Specifically, we use the sequence of answers in Table 1 to establish a mapping. The QA pair template is {Q: "The solutions of (x-a)(x-b)(x-c)(x-d)(x-e)=0 are x=", A: xxx} and each QA pair can be embedded with 5! = 120-bit watermarks.

As shown in Table 9, we can embed a 120-bit watermark with an ESR that exceeds 90%. Even if the watermark capacity reaches 240 bits, the ESR can still exceed 75%. These results demonstrate that our watermarking method is scalable. We can choose different types of mapping to embed watermarks according to the requirements of watermark capacity.

| Watermark Capacity | GPT-J-6B | | Baichuan-7B | | LLaMA-7B | |
|---|---|---|---|---|---|---|
| | ESR | ET | ESR | ET | ESR | ET |
| 120bit | 100.0% | 72.3s | 90.0% | 66.4s | 90.0% | 58.6s |
| 240bit | 100.0% | 148.3s | 78.3% | 98.5s | 75.0% | 83.7s |

Table 9: The performance of our watermarking method using the watermarked mapping based on the sequence of answers.

*3) Model Editing Technique.* To evaluate the scalability of our watermarking method on the model editing technique, we select EMMET Yoon et al. (2024) as the model editing technique to embed the watermark into LLM.

As shown in Table 10, the results indicate that the ESR exceeds 95% when embedding 8-bit and 16-bit watermarks, which demonstrates the effectiveness of our watermarking method when using EMMET. Therefore, our watermarking method is scalable on the model editing technique that supports the editing of multiple knowledge instances.

| Watermark Capacity | GPT-J-6B | | GPT2-XL | | LLaMA-7B | |
|---|---|---|---|---|---|---|
| | ESR | ET | ESR | ET | ESR | ET |
| 8bit | 100.0% | 112.3s | 100.0% | 49.4s | 95.0% | 131.2s |
| 16bit | 100.0% | 176.1s | 100.0% | 53.2s | 99.2% | 194.6s |

Table 10: The performance of our watermarking method using the EMMET to embed watermark.

# 6 CONCLUSION

In this paper, we introduce EditMark, a novel training-free and harmless watermarking method for open-source Large Language Models. By leveraging the inherent diversity in LLM responses, Edit-Mark embeds watermarks through model editing without affecting the logical and semantic integrity of the generated text. Our method demonstrates the effectiveness, robustness, fidelity, and efficiency of EditMark, marking a significant advancement in LLM watermarking and providing a practical solution for protecting model copyrights. We hope EditMark could be a feasible watermarking method for open-source LLMs to protect their copyright and prevent unauthorized use or resale.

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

# A APPENDIX

## A.1 IMPLEMENTATION DETAILS.

In the following, we will introduce the specific experimental settings of our watermarking method. Specifically, we select *MEMIT* as the model editing technique to embed the watermark. The template of the QA pair we embed the watermark is {Q: "The value of xxx/xxx is", A: "xxx"} where "The value of xxx/xxx is", "xxx/xxx", and "xxx" are the input, subject, and target output during the process of model editing, respectively. For experiments with different watermark capacities, we performed five independent experiments to calculate the average ESR and embedding time, where the random seeds for the five independent experiments were 1, 2, 3, 4, and 5. For the hyperparameter of inference, the temperature is 0.0, and max_tokens is 10. The hyperparameters for model editing are set as follows: $\alpha = 2$, $\gamma = 3$, $m = 2$, $\beta = 2$, and a maximum of editing round $N = 6$. Each LLM is edited for at least two epochs and stops after the second epoch when all QA pairs are successfully edited. We modify the weights of MLP layers to embed the watermark. And the MLP layers we edit are "9, 10, 11, 12, 13, 14" for LLaMA-7B, "13, 14, 15, 16, 17" for GPT2-XL, "3, 4, 5, 6, 7, 8, 9" for GPT-J-6B and "4, 5, 6, 7, 8, 9" for "Baichuan-7B". We used two RTX 4090 24GB to complete the model editing, and all the LLMs are loaded to float16.

## A.2 WATERMARK EMBEDDING ALGORITHM.

The following is the pseudo-code for QA pairs generation and watermark embedding, which is detailed in Algorithm 1.

---

**Algorithm 1:** QA Pairs Generation and Watermark Embedding

**Input**  : Watermark: $\mathcal{W} = \{0, 1\}^n$, Grouping parameter: $m$, Number of bits: $n$, Secure key: $S$
         Hyperparameters: $\alpha, \beta, \gamma$.
**Output:** Questions: $Q$, Answers: $A$.

1 Initialize the encoded watermark: $\mathcal{W}' = \{\}$;
2 **for** $i = 1$ *to* $n/m$ **do**
3     $\omega_i = \{\}$;
4     **for** $j = 1$ *to* $m$ **do**
5          Add $\mathcal{W}[i * m + j]$ to $\omega_i$; *Group the watermark.*;
6     **end**
7     $\omega_i' = (\omega_i)_{10} + \alpha$;   *Encode $\omega_i$ to the corresponding decimal number and add $\alpha$*;
8     Add $\omega_i'$ to $\mathcal{W}'$;
9 **end**
10 Use the secure key to generate the pseudo-random number sequence $t$;
11 $t = \{t_0, t_1^1, t_1^2, \ldots, t_1^\gamma, t_2^1, t_2^2, \ldots, t_k^{\gamma-1}, t_k^\gamma\}$ *s.t.* $\forall x, y \in \mathbf{t}, \mathbf{x} \neq \mathbf{y}$.;
12 Generate questions and answers of QA pairs.;
13 Initialize Questions: $Q = \{\}$ ;
14 Initialize Answers: $A = \{\}$ ;
15 **for** $i = 1$ *to* $(n/m)$ **do**
16     **for** $j = 1$ *to* $\gamma$ **do**
17         Add the question "The value of $t_i^j/t_0$ is" to $Q$;
18         Calculate the value of $t_i^j/t_0$ and keep $\mathcal{W}_i'$ decimal places to obtain the answer: $a$;
19         Add the answer "$a$." to $A$;
20     **end**
21
22 **end**
23 **return** Questions: $Q$, Answers: $A$;

---

## A.3 WATERMARK EXTRACTION ALGORITHM.

The following is the pseudo-code for extracting the watermark from QA pairs, which is detailed in Algorithm 2.

---

**Algorithm 2:** Watermark Extraction from QA Pairs

---

**Input** : Question: $Q$, Answer: $A$, Grouping parameter: $m$, Number of bits: $n$, Secure key: $S$
         Hyperparameters: $\alpha$, $\beta$, $\gamma$.
**Output:** Watermark: $\mathcal{W}$

1 Initialize watermark: $\mathcal{W} = \{\}$;
2 Extract the encoded watermark from the Questions and Answers.;
3 **for** $i = 1$ *to* $(n/m)$ **do**
4     Initialize $\widetilde{\omega}_i = \{\}$;
5     **for** $j = 1$ *to* $\gamma$ **do**
6        Extracting the dividend $d$ in a question $Q[i * \gamma + j]$;
7        Extracting the precision $p$ of the answer $A[i * \gamma + j]$;
8        $\widetilde{\omega}_i^j = p - d \bmod \beta$;
9        Add $\widetilde{\omega}_i^j$ into $\widetilde{\omega}_i$;
10     **end**
11     $\omega_i' = \text{vote}(\widetilde{\omega}_i)$; *Calculate the majority of $\widetilde{\omega}_i$;*
12     $\omega_i = (\omega_i' - \alpha)_2$; *Decode the decimal number $\omega_i'$ into the corresponding binary number $\omega_i$;*
13     **for** $j = 1$ *to* $m$ **do**
14        Add $\omega_i^j$ to $\mathcal{W}$;
15     **end**
16 **end**
17 **return** Watermark: $\mathcal{W}$;

---