# OpenReview forum: "EditMark: Training-free and Harmless Watermark for Large Language Models"
_ICLR.cc/2025/Conference — Submitted to ICLR 2025_

### Official Review · Reviewer_sRPi · 2024-11-03

**Soundness:** 3
**Presentation:** 4
**Contribution:** 2
**Rating:** 5
**Confidence:** 3

**Summary:**

This paper presents a black-box watermarking method for large language models that is both training-free and harmless. The work addresses the scenario of copyright protection for large language models by applying a model editing technique to embed the watermark.

**Strengths:**

+）The writing is clear and easy to understand.

+）The authors propose a new watermarking mechanism for large language models.

**Weaknesses:**

-）Although the paper applies a model editing technique, the technical contribution is limited.

-）The method used for editing the model is not properly cited. It's unclear which paper is referred to as "Memit" (lines 153-154).
Providing more examples would improve the paper.

-）The baseline selection is somewhat limited, as seen with the choice of "A watermark for large language models" [1].

-) If the attacker changes the sampling strategy (e.g., adjusts the temperature of the softmax layer), can the watermark maintain its robustness? The authors may consider extending the experiments.

[1] Kirchenbauer J, Geiping J, Wen Y, et al. A watermark for large language models[C]//International Conference on Machine Learning. PMLR, 2023: 17061-17084.

**Questions:**

Please see weakness.

---

> ### Author Response · Authors · 2024-11-25
>
> Thank you for your valuable and constructive comments.
>
> **1.Weekness**
>
> **Re1-1** Thank you for your valuable feedback. We would like to clarify the core contribution of our paper. Unlike the efforts of the study that aim to propose new technologies, such as introducing new LLMs and novel training methods, the goal of our paper is to design a train-free and harmless watermark embedding framework. While we build on existing techniques, our work introduces other contributions, such as using model editing to embed watermarks into LLM for the first time and designing a harmless watermark embedding method that guides the responses to open-ended questions by leveraging the diversity of LLM responses.  These are important advancements in harmless and train-free watermarks for LLMs, and the experimental results also verify our contribution.
>
> Like the existing backdoor-based watermarking method[1], although it uses established training technologies, it was groundbreaking to apply backdoors to watermarking for the first time. Similarly, while our watermarking method leverages existing techniques, it offers contributions by adapting these technologies to create a harmless and efficient watermark framework. Therefore, we sincerely hope that you will pay more attention to the contributions of our watermarking framework based on model editing.
>
> **Re1-2** Thank you for your correction. The corresponding paper for Memit is [2]. We have corrected this citation in the revised paper.
>
> **Re-1-3** The watermarking method[3] you mentioned is embedding watermarks into the generated text, which has been widely used to identify AI-generated text. Specifically, this method involves defining green and red token sets and modifying the logits to make the LLM basis generate green tokens.
>
> However, this watermarking method is not suitable for protecting the copyright of the open-source LLMs.  This is because this watermarking method requires modifying logits to make LLM bias to sample green tokens during inference, which requires additional codes to achieve this. However, for open-source LLMs, attackers can access LLMs in a white-box and control the inference process. Experienced attackers can easily find these codes for the watermarking process and remove them to remove the watermark. Therefore, we did not select this method as a baseline for comparison.
>
> **Re-1-4** Thank you for your valuable feedback. As shown in Re-Table 1, we also evaluated the effectiveness of our method when the temperature is 1.0. The results show that the ESR of our watermarking method is also close to 100% when the temperature is 1.0, which validates that our watermarking method is effective when changing the sampling strategy (adjusting the temperature).
>
> Re-Table 1: The ESR of our watermarking method when the temperature is 1.0.
>
> | Watermark Capactiy | GPT-J-6B | GPT2-XL | Baichuan-7B | LLaMA-7B |
> | :----------------: | :------: | :-----: | :---------: | :------: |
> |        8bit        |  100.0%  |  98.3%  |   100.0%    |  95.0%   |
> |       16bit        |  100.0%  |  94.2%  |    99.2%    |  96.7%   |
>
>
>
> [1] Adi Y, Baum C, Cisse M, et al. Turning your weakness into a strength: Watermarking deep neural networks by backdooring[C]//27th USENIX security symposium (USENIX Security 18). 2018: 1615-1631.
>
> [2] Meng K, Sharma A S, Andonian A J, et al. Mass-Editing Memory in a Transformer[C]//The Eleventh International Conference on Learning Representations.
>
> [3] Kirchenbauer J, Geiping J, Wen Y, et al. A watermark for large language models[C]//International Conference on Machine Learning. PMLR, 2023: 17061-17084.

---

> > ### Comment · Reviewer_sRPi · 2024-12-03
> >
> > Thank you for your response, which addresses my concerns. By referencing your conversation with other reviewers, I decide to maintain my score

---

### Official Review · Reviewer_hcYM · 2024-11-03

**Soundness:** 3
**Presentation:** 2
**Contribution:** 2
**Rating:** 3
**Confidence:** 3

**Summary:**

The paper introduces EditMark, a novel method for watermarking large language models (LLMs) that is both training-free and harmless. The core idea behind EditMark is to leverage the inherent diversity in LLM responses to open-ended questions to embed watermarks without degrading the model's performance. The method involves generating a harmless mapping based on the watermark, selecting a secret key to generate watermarked inputs, and editing the outputs of the LLM to align with the harmless mapping. The authors claim that EditMark can embed 8-bit watermarks into LLMs within 2 minutes, with a watermark extraction success rate close to 100%. The method is also robust to model fine-tuning and editing attacks, maintaining high fidelity across various benchmarks.

**Strengths:**

1. **Training-Free**: One of the most significant strengths of EditMark is that it does not require retraining the LLMs. This makes it highly efficient and cost-effective, especially for large-scale models.
2. **Harmlessness**: The method ensures that the watermark does not affect the logical and semantic integrity of the generated text, which is crucial for maintaining the model's performance and usability.
3. **Efficiency**: The authors report that EditMark can embed watermarks quickly, within 2 minutes, making it a practical solution for real-world applications.
4. **Robustness**: The method is robust against various attacks, including model fine-tuning and editing attacks, which enhances its reliability.
5. **Fidelity**: Extensive experiments on benchmarks such as BLiMP and MMLU show that EditMark has a negligible effect on model performance, indicating high fidelity.

**Weaknesses:**

1. **Complexity of Implementation**: While the method is training-free, the process of generating a harmless mapping and selecting a secret key might be complex and require sophisticated algorithms. The paper could benefit from more detailed explanations and examples of these processes.
2. **Scalability**: The paper mentions that EditMark can embed 8-bit watermarks, but it is unclear how well the method scales to larger watermarks. Embedding more bits might introduce additional challenges and potential degradation in performance.
3. **Security Concerns**: Although the method is designed to be harmless, the security of the watermarking process is not thoroughly discussed. There is a need to explore potential vulnerabilities and countermeasures against advanced attackers who might attempt to reverse-engineer the watermarking mechanism.
4. **Generalizability**: The paper primarily focuses on open-source LLMs, but it is not clear how well EditMark would perform on proprietary models with different architectures and training methodologies. More diverse experimental settings would strengthen the paper's claims.
5. **User Impact**: The paper does not discuss the potential impact of the watermark on user experience. For example, if the watermark alters the model's responses in subtle ways, it could affect the trust users place in the model's outputs.

**Questions:**

1. **Detailed Implementation**: Could the authors provide more detailed steps and pseudocode for generating the harmless mapping and selecting the secret key? How do these steps ensure the watermark remains undetectable?
2. **Scalability**: How does the method scale to larger watermarks? Are there any limitations or trade-offs when embedding more bits?
3. **Security**: What are the potential security risks associated with the watermarking process, and how can they be mitigated? Have the authors conducted any security audits or simulations to test the robustness of the method against advanced attacks?
4. **Generalizability**: How well does EditMark perform on proprietary models with different architectures? Are there any modifications needed to adapt the method to non-open-source LLMs?
5. **User Experience**: How does the watermark affect the user experience? Are there any noticeable changes in the model's behavior that could impact user trust?

---

> ### Author Response · Authors · 2024-11-25
>
> Thank you for your valuable and constructive comments.
>
> **1.Weekness**
>
> **Re1-1** Generally, our watermark embedding process is not complicated and easy to implement. The algorithm's complexity does not affect the performance and deployment of the watermark. To help you better understand the process of generating QA pairs and embedding watermarks, we provide the Python code below.  However, we appreciate your insightful comments and have presented pseudocode for these algorithms in the revised paper to help readers better understand our watermark embedding and extraction process.
>
> ```python
> import random
> from collections import Counter
>
> bit = 8
> watermark = [random.choice([0, 1]) for i in range(bit)]
> gama,m,alpha,beta,key = 3,2,2,2,1
>
>
> def generate_QA(bit,watermark,gama,m,alpha,beta,key):
>     w = [int(''.join(map(str, watermark[i*m:i*m+m])), 2) + alpha for i in range(bit//m)]
>     random.seed(key)
>     divisor, S = random.randint(50, 100), random.sample(range(1, 501), (bit//2)*3)
>     Q = [f'The value of {j}/{divisor} is' for j in S]
>     def divide(a,b,precision):
>         c = str(a/b)
>         return c[:c.find('.')+precision+1]
>     A = ['%s.'%(divide(s,divisor,s % beta + w[i//gama])) for i, s in enumerate(S)]
>     return Q, A, S
>
> def extract_watermark(S, answers, m, beta, alpha, gama):
>     W = []
>     for i in range(0,len(sequence),gama):
>         w = []
>         for j in range(gama):
>             p = len(answers[i+j].split('.')[1])
>             w.append(p - S[i+j] % beta)
>         e_w = Counter(w).most_common(1)[0][0] - alpha
>         decoded_w = bin(e_w)[2:].zfill(m)[-m:]
>         for j in range(m):
>             W.append(decoded_w[j])
>     return W
>
> prompts, answers, sequence = generate_QA(bit,watermark,gama,m,alpha,beta,key)
> extract_watermark = extract_watermark(sequence, answers, m, beta, alpha, gama)
> ```
> **Re1-2** **We have already presented the results and analysis of ESR under different watermark capacities in Section 5.3.**  As shown in Table 3, when embedding more bits of the watermark (16-bit, 32-bit, and 48-bit), the ESR does not decrease significantly and is close to 100%. These results demonstrate that our watermarking method has good scalability when embedding multi-bit watermarks.
>
> **Re1-3** We guess that the security you mentioned might be robustness against adaptive attacks. For advanced attackers, we have already analyzed adaptive attacks in Section 5.4. We assume that the attacker knows the watermarking method and the parameters of the embedded watermark. We acknowledge that if attackers know the specific QA pairs used for embedding the watermark, they could potentially remove it. However, it is challenging for attackers to identify the specific QA pairs used for watermark embedding. As shown in Table 5 and Figure 3,  when the attacker is unaware of the QA pairs, it is difficult to remove the watermark through fine-tuning or model editing attacks.  These results demonstrate that our method remains robust against adaptive attacks in real-world scenarios. In addition,  we can design multiple templates to generate QA pairs and edit them to embed watermarks to enhance their robustness.
>
> **Re1-4** Open-source LLM refers to LLM that can be accessed by attackers in a white-box scenario, which includes general LLM and proprietary LLM.  In addition, the LLMs we selected, such as GPT2-XL, GPT-J-6B, and LLaMA-7B, have different architectures and pre-training processes. Following your advice, we will add some proprietary LLMs in the revised paper.
>
> **Re1-5** Our watermarking method has little impact on the user experience. Specifically, the questions in the QA pairs have multiple correct answers, and we use the watermark to control LLM to select a correct answer as a response, which means that the QA pairs are factually and logically correct.  Second, according to the characteristics of model editing technology, there is almost no change in the output of the content that is not edited. Therefore, users can hardly feel the difference between watermarked and non-watermarked models.
>
>
>
> **2. Question**
>
> **Re2-1** We have presented the codes for QA pairs generation and extracting watermark in **Re1-1**. In addition, We do not claim that our watermark is undetectable.  However, our watermark is difficult to detect since the QA pairs to embed the watermark are logically and factually correct.
>
> **Re2-2** We have already presented the results of ESR under different capacities and analyzed the trade-off in Section 5.3.
>
> **Re2-3** We answered this question in Re1-3.
>
> **Re2-4** The LLMs we selected, such as GPT2-XL, GPT-J-6B and LLaMA-7B, are LLMs with different architectures. The non-open-source LLM you mentioned refers to the LLM API or proprietary LLM?  If it is for LLM API or proprietary LLM, our watermarking method does not need to be modified and adapted since it is generalizable.
>
> **Re2-5** Our watermarking method has little impact on the user experience.  There are no noticeable changes in the LLM's behavior.

---

### Official Review · Reviewer_HcpJ · 2024-11-03

**Soundness:** 3
**Presentation:** 2
**Contribution:** 3
**Rating:** 6
**Confidence:** 3

**Summary:**

The paper introduces EditMark, a novel approach for watermarking open-source large language models (LLMs) that is both training-free and harmless. Unlike existing methods that rely on backdoors or knowledge injection, EditMark leverages model editing to embed watermarks without significant alterations to the model’s output quality.

**Strengths:**

1. The EditMark method introduces a unique, training-free watermarking solution leveraging model editing, avoiding the significant computational costs and potential risks associated with backdoor and knowledge injection methods. This innovation provides a practical means to protect open-source LLMs.

2. The proposed method can embed an 8-bit watermark within approximately two minutes, as demonstrated across multiple LLMs. This rapid embedding process, combined with a high extraction success rate (ESR) close to 100%, highlights the method's effectiveness and time-efficiency compared to existing techniques.

3. Experiments conducted using the BLiMP and MMLU benchmarks show that EditMark has a negligible impact on the model’s overall performance. The watermarking process preserves the logical and semantic integrity of the LLM’s output, ensuring that model functionality is maintained.

4. EditMark demonstrates notable resilience to common model attacks, including fine-tuning and editing. The watermark extraction success rate remains high even after such modifications, indicating that EditMark's approach offers robust protection without significantly compromising model usability.

**Weaknesses:**

1. The paper mentions the trade-off between watermark capacity, ESR, and embedding time. Although embedding an 8-bit watermark is efficient and robust, scalability to higher capacities could present challenges, potentially limiting its utility in scenarios requiring more extensive watermarking.

2. The robustness of EditMark against an informed attacker (i.e., one who knows the specific model editing layers used) shows some vulnerabilities. As depicted in experiments with targeted attacks, the watermark's resilience could be compromised, suggesting that further enhancement in adaptability is needed.

3. The reliance on the Memit model editing technique limits the flexibility of the approach. While Memit is efficient for editing multiple knowledge instances, the method’s compatibility with other editing frameworks is not addressed, which could be a limitation for broader application.

4. The paper's citation formatting is problematic. The authors may use "(authors et al.)" to highlight the citation part rather than directly show "authors et al."

**Questions:**

1. How scalable is the EditMark framework when embedding larger watermarks (e.g., 16-bit or 32-bit) without significantly affecting ESR and embedding time?

---

> ### Author Response · Authors · 2024-11-25
>
> Thank you for your valuable and constructive comments.
>
> **1.Weekness**
>
> **Re1-1** We admit that increasing the watermark capacity will reduce the ESR since the knowledge of LLM that can be edited at one time by model editing technology is limited. However, this issue can be addressed by using other QA pairs to embed the watermark. For example, as shown in Table 1, we can use the diversity of the order of LLM's answers to design QA pairs for embedding watermarks. Assuming the question is `'{Q: The solutions of (x-a)(x-b)(x-c)(x-d)(x-e)=0 are x=, A:xxx}.`' For this question, there are **5! = 120** correct answers according to the order of answers. For example, for this question: `'The solutions (x-1)(x-2)(x-3)(x-4)(x-5)=0 are x=,`' `'1,2,3,4`,5' and `'1,3,2,4,5`' are all correct answers. Therefore, each QA pair can be embedded with **5! = 120** bit watermarks. Similarly, if the question is `'(x-1)(x-2)...(x-n)=0`', then there will be **n!** correct answers, which means that editing a QA pair can embed **n!** bits of the watermark.
>
> We also evaluate the effectiveness of our watermarking method using these QA pairs to embed watermarks. As shown in Re-Table 1, we can embed a 120-bit watermark with an ESR that exceeds 90%. Even if the watermark capacity reaches 240 bits, the ESR can still exceed 75%. The results indicate that although our method has a trade-off in ESR and watermark capacity, it can still meet the needs of large-capacity watermark embedding.
>
>
>
> Re-Table 1: The ESR of using the QA pairs based on order to embed watermarks.
>
> | Watermark Capacity | GPT-J-6B | Baichuan-7B | LLaMA-7B |
> | :----------------: | :------: | :---------: | :------: |
> |       120bit       |  100.0%  |    90.0%    |  90.0%   |
> |       240bit       |  100.0%  |    78.3%    |  75.0%   |
>
> **Re1-2** We appreciate your insightful comments on the robustness of EditMark against adaptive attacks. We acknowledge that if attackers know the specific QA pairs used for embedding the watermark, they could potentially leverage model editing attacks to remove it.  In real-world scenarios, the attacker can know the watermarking method we use and know the parameters used to embed the watermark. However, it is challenging for attackers to identify the specific QA pairs used for watermark embedding. As shown in Table 5 and Figure 3,  when the attacker is unaware of the QA pairs, it is difficult to remove the watermark through fine-tuning or model editing attacks.  These results demonstrate that our method remains robust against adaptive attacks in real-world scenarios.
>
> In addition,  we can design multiple templates to generate QA pairs and edit them to embed watermarks to enhance their robustness against adaptive attacks. In this way, even if the attacker knows that we use one or a set of QA pairs to embed watermarks and remove these watermarks, other types of QA pairs are still effective, which can enhance the robustness of our watermarking method against adaptive attacks.
>
> **Re1-3** We would like to clarify that MEMIT was chosen as a representative example to demonstrate the feasibility and effectiveness of our watermarking method. In addition, our watermarking method is not inherently tied to MEMIT, and it can be extended to other model editing frameworks that support the editing of multiple knowledge instances. For example,  model editing techniques like EMMET could also be used to implement our method. As shown in Re-Table 2, we have evaluated the effectiveness of our watermarking method when using EMMET[1] to embed watermarks. The results indicate that the ESR exceeds 95%, which demonstrates the effectiveness of our watermarking method when using EMMET.
>
> Re-Table 2: The ESR of using the EMMET to embed the watermark.
>
> | Watermark Capacity | GPT-J-6B | GPT2-XL | LLaMA-7b |
> | :----------------: | :------: | :-----: | :------: |
> |        8bit        |  100.0%  | 100.0%  |  95.0%   |
> |       16bit        |  100.0%  | 100.0%  |  99.2%   |
>
> **Re1-4** Thank you for pointing out our citation problem. Following your advice, we have revised the citation format in the revised paper to make the citation format standard.
>
> [1] Yoon J, Gupta A, Anumanchipalli G. Is Bigger Edit Batch Size Always Better?--An Empirical Study on Model Editing with Llama-3[J]. arXiv preprint arXiv:2405.00664, 2024.
>
> **2.Question**
>
> **Re2-1**  We have already presented the results and analysis of ESR and watermark embedding time under different watermark capacities in Section 5.3. The results are detailed in Table 3 in the paper. As shown in Table 3, when embedding more bits of the watermark (16-bit, 32-bit, and 48-bit), the ESR does not decrease significantly and is close to 100%. In addition, the embedding time also does not increase significantly. The increase in embedding time increases linearly with the increase in watermark capacity. These results demonstrate that our watermarking method has good scalability when embedding multi-bit watermarks.

---

### Official Review · Reviewer_7QD4 · 2024-11-04

**Soundness:** 2
**Presentation:** 2
**Contribution:** 2
**Rating:** 3
**Confidence:** 3

**Summary:**

The paper tackles the issues of watermarking LLMs, such that a LLM provider can track the usage of its models by third parties. Backdoors or knowledge injections have drawbacks. Instead, EditMark propose to align with a harmless watermark mapping to open-ended questions, such as controling the number of digits when generating a float.

**Strengths:**

The topic is important, and adequately using knowledge injection to watermark LLMs is interesting. Doing so with open-ended questions, particularly with math questions, seem relevant and adequate.

**Weaknesses:**

- The paper has a lot of flows. A lot of paragraphs really look LLM written and hard to digest
- The fact that the method is simple is a good thing. However, the explanation of the watermark extraction and experimental set-up appears very light. It would help to add in the appendix more details on the experiments, examples, related work. Overall I am not convinced by the literature review, and the choice of methods that the authors compare to. It is not clear wheter the novelty comes from the choice of watermarking or the choice of doing model editing instead of full finetuning.

Overall, the method is interesting but the paper is too drafty.



Minor:
- The LLMs that are considered are a few years old, I don't understand why.
- The authors could use \citep insstead of cite/citet when adequate.
- "Kirchenbauer et al. proposed the first watermarking methods Kirchenbauer
et al. (2023) for LLM." What about aaranson et al ?

**Questions:**

When reading this (which has flaws):

"To address this issue, Li et al. proposed a watermarking method Li et al. (2024) via knowledge injection, which embeds the watermark into the knowledge and injects the watermarked knowledge into LLM. In addition, the watermarked text is logically correct, which makes it harmless for LLM. However, this method also requires training the LLMs to embed watermarks, which is a huge cost for large-size LLMs.""


- I understand that there are already some work to insert watermarked harmless knowledge into a LLM. But it needs full fine-tuning so the authors don't compare to them. On the other hand the proposed method uses model editing. But couldn't that be applied to the previous harmless knowledge injection approached?

- Does the novelty come from the proposed method of watermarking the ouputs of math questions, or from the model editing part?

- Could the authors clarify how detection is done?

- What is the field "begin" in the table?

- Could the authors detail how they generated the question-answer pairs? how many of them are used for training? how many are used for detection? Are they the same questions as the training ones?

- In figure 3, why is Llama not as good as the other models?

---

> ### Author Response · Authors · 2024-11-25
>
> Thank you for your valuable and constructive comments.
>
> **1. Weakness**
>
> **Re1-1** We have revised the citation format in the revised paper to make the citation format standard.  We admit that we used LLM to correct grammatical errors.  We have revised our papers to help you better understand our paper.
>
> **Re1-2** Firstly, We would like to clarify the novelty of our watermarking method, which stems from both the development of a training-free and harmless method for embedding watermarks. Specifically, this is the first instance where multi-bit, harmless watermarks have been embedded using model editing techniques. Compared with the harmless knowledge injection method based on fine-turning, our method is more suitable for model editing since the edited content is brief and short. In addition, for the harmless part, we use watermark context to guide the responses to open-ended questions by leveraging the diversity of LLM responses. Because large models have many alternative responses to many open questions, we inject watermarks into this response redundancy. Consequently, the watermarked text is only a slight modification of the natural text and is guaranteed to be logically and factually correct, making it more covert.
>
> Additionally, we have added an appendix to introduce the experimental settings and the pseudocode for both the watermark embedding and extraction in the revised paper, and the following is their Python code.
>
> ```python
> import random
> from collections import Counter
>
> bit = 8
> watermark = [random.choice([0, 1]) for i in range(bit)]
> gama,m,alpha,beta,key = 3,2,2,2,1
>
>
> def generate_QA(bit,watermark,gama,m,alpha,beta,key):
>     w = [int(''.join(map(str, watermark[i*m:i*m+m])), 2) + alpha for i in range(bit//m)]
>     random.seed(key)
>     divisor, S = random.randint(50, 100), random.sample(range(1, 501), (bit//2)*3)
>     Q = [f'The value of {j}/{divisor} is' for j in S]
>     def divide(a,b,precision):
>         c = str(a/b)
>         return c[:c.find('.')+precision+1]
>     A = ['%s.'%(divide(s,divisor,s % beta + w[i//gama])) for i, s in enumerate(S)]
>     return Q, A, S
>
> def extract_watermark(S, answers, m, beta, alpha, gama):
>     W = []
>     for i in range(0,len(sequence),gama):
>         w = []
>         for j in range(gama):
>             p = len(answers[i+j].split('.')[1])
>             w.append(p - S[i+j] % beta)
>         e_w = Counter(w).most_common(1)[0][0] - alpha
>         decoded_w = bin(e_w)[2:].zfill(m)[-m:]
>         for j in range(m):
>             W.append(decoded_w[j])
>     return W
>
> prompts, answers, sequence = generate_QA(bit,watermark,gama,m,alpha,beta,key)
> extract_watermark = extract_watermark(sequence, answers, m, beta, alpha, gama)
> ```
> **2. Minor**
>
> **Re2-1** GPT-J-6B and GPT2-XL, LLaMA-7B are commonly used LLMs in the existing research about model editing.
>
> **Re2-2** We have revised the citations based on your advice and used \citep to standardize the citation format.
>
> **Re2-3**  We have revised the related work based on your comments.
>
> **3. Question**
>
> **Re3-1** Your analysis is correct. The watermarking method based on knowledge injection is harmless. However, using the model editing technique to inject watermarked knowledge into LLM is not effective enough compared with fine-tuning since it needs to inject a variety of complex and long watermarked knowledge, which limits its effectiveness.
>
> **Re3-2**  We clarify the novelty of our watermarking method in Re1-2, which comes from train-free and harmless.
>
> **Re3-3** We will present an example of watermark extraction in the following.
>
> 1. Set the gama = 3, m=2, alpha=2, beta = 2, and secret key = 1.
> 2. Use the secret key to generate the divisor (e.g., 58) and the random sequence (e.g., [292, 434, 411,...]).
> 3. Generate the questions based on the sequence: ["The value of 292/58 is",...]
> 4. Query the target LLM and obtain the processed answers:  ['5.0344.', '7.4827.', '7.08620.',...].
> 5. We take the first group as an example. Since the gama=3, the first group of answers is  ['5.0344.', '7.4827.', '7.08620.']. Since the dividend in the first question is 292, and the precision in the first answer is 4, the first encoded watermark is $w_0^0=4-292 ~ mod ~ \beta=4$. Similarly, we can determine that $w_0^1=4$ and $w_0^2=4$. Therefore, we can vote that the encoded watermark in the first group is $w_0=4$. Finally, the watermark in the first group can be decoded: $(w_0-\alpha)_2=01$.
>
> **Re3-4** The ``begin'' in Table 5 represents the original watermarked LLMs, and we have revised it in the revised paper.
>
> **Re3-5** We have presented the codes for QA pairs generation in Re1-2. The number of QA pairs is already detailed in Table 3. The questions used for embedding and extracting watermarks are the same.
>
> **Re3-6**  Memit locates the neurons in the MLP layer and then edits them. For LLaMA-7B, the location of the edited knowledge in the neurons may not be as accurate as other LLMs during model editing. Therefore, its ESR is lower than that of other LLMs.

---

### Meta-Review · Area_Chair_ciik · 2024-12-16

**Metareview:**

This paper introduces EditMark, a training-free and harmless method for watermarking large language models (LLMs) using model editing. Unlike existing approaches that rely on backdoors or knowledge injection, EditMark embeds watermarks by leveraging the natural diversity in LLM outputs. The method creates a harmless mapping, generates watermarked inputs using a secret key, and edits model outputs to match the mapping. EditMark can embed 8-bit watermarks in under 2 minutes, achieving near 100% extraction success. It is robust to fine-tuning and editing attacks, ensuring high fidelity and no degradation in model performance, making it suitable for copyright protection.

The idea of inserting information without training the LLM is an interesting direction. However, as noted by the reviewers, the comparison to existing methods is limited. Since LLM watermarking is a hot topic, there are numerous existing works [1]. Therefore, it would be beneficial to discuss the differences between the proposed method and existing methods, and it would be great to include some additional comparisons. I encourage the authors to revise the paper based on the reviewers' comments and resubmit it to a future venue.

[1] A Survey of Text Watermarking in the Era of Large Language Models
https://dl.acm.org/doi/full/10.1145/3691626

**Additional Comments On Reviewer Discussion:**

The authors responded to the concerns raised by the reviewers. The reviewers highlighted issues with the paper’s writing, literature survey, and comparison to existing baselines. While the response addressed some concerns, the average score remains below the borderline, and one reviewer remains unenthusiastic after the rebuttal. Overall, I agree that the literature survey is insufficient and that the paper requires major revisions. Therefore, I recommend rejecting this paper.

---

### Decision · Program_Chairs · 2025-01-22

Reject